# The Effects of Cognitive Behavioral Therapy for Insomnia among College Students with Irritable Bowel Syndrome: A Randomized Controlled Trial

**DOI:** 10.3390/ijerph192114174

**Published:** 2022-10-29

**Authors:** Yun-Yi Yang, Sangeun Jun

**Affiliations:** 1Department of Nursing, Healthcare Science & Human Ecology, Dong-Eui University, Busan 47340, Korea; 2College of Nursing, Keimyung University, Daegu 42601, Korea

**Keywords:** irritable bowel syndrome, cognitive therapy, sleep, gastrointestinal

## Abstract

The aim of the study was to develop and evaluate cognitive behavioral therapy for insomnia (CBT-I) among college students with irritable bowel syndrome (IBS). We randomly assigned 60 college students with IBS comorbid insomnia to the experimental group who received CBT-I for 90 min once a week for 4 weeks and the control (non-CBT-I) group. Participants completed self-report measures of insomnia severity, pre-sleep arousal, sleep-related dysfunctional cognitions, maladaptive sleep habits, IBS symptom severity and IBS quality of life (QOL) at baseline, after intervention, and at 3-month follow-up. Sleep pattern, GI symptoms during sleep and Interleukin-6 (IL-6) and C-Reaction Protein (CRP) were measured at baseline and after intervention. The experimental group showed significant decreases in insomnia severity, sleep onset latency, total time in bed, pre-sleep arousal, GI symptoms during sleep, sleep-related dysfunctional cognitions, maladaptive sleep habits, and IBS symptom severity, compared with the control group. This group also showed significant increases in sleep efficiency and IBS QOL compared with the control group. No significant differences were observed between the levels of IL-6 and CRP of both groups. CBT-I for college students with comorbid IBS and insomnia was effective in reducing insomnia, IBS symptom severity, and IBS QOL.

## 1. Introduction

Irritable bowel syndrome (IBS) is a functional disorder in the gastrointestinal (GI) tract (Drossman, 2006) [1]. It is often characterized by recurrent abdominal pain or discomfort associated with altered bowel pattern (i.e., constipation, diarrhea, or mixed diarrhea-constipation) [1]. About 11% of the world’s population have IBS [2], and its prevalence among college students is 10.7~35.5% [3,4], which is higher than other age groups. Insomnia is a comorbid disorder commonly seen in IBS patients [5] and there are reports that 53.3% of college students with IBS are affected by insomnia [6]. Although the mechanism for the comorbidity of IBS and insomnia has not been clearly identified, it is speculated that they are connected by the gut-brain axis [7]. Several studies suggest that sleep problems could occur because of gastrointestinal (GI) symptoms, given the increased hypersensitivity to visceral pain due to the imbalance of the autonomic nervous system (ANS) and hypothalamic pituitary adrenal axis (HAP-axis) [8,9,10]. Furthermore, insomnia could remain even after the comorbid disease is cured [11].

Insomnia, thus, becomes a source of stress and leads to the increase of inflammatory reactions [12] and the recurrence of the comorbid disease, lowering the quality of life [11]. Therefore, interventions for comorbid IBS with insomnia should also focus on insomnia [13]. However, a study, in which melatonin, a sleep promoting agent, was administered to IBS patients with insomnia, found that the medication did not improve sleep [14]. Contrarily, two studies that included sleep hygiene education through comprehensive self-management intervention program, showed significant improvement in IBS symptoms [15,16]. Unfortunately, these studies did not assess the impact of the intervention programs for sleep improvement.

An effective intervention for treating insomnia, recommended by the American Academy of Sleep Medicine, is cognitive behavioral therapy for insomnia (CBT-I) [17]. Based on the Spielman’s chronic insomnia 3-P model, the focus of the CBT-I are the perpetuating factors responsible for insomnia so that distorted cognitions and behaviors can be remedied [18]. Furthermore, several studies reported that it was helpful to include interventions that address the symptoms and characteristics of a comorbid condition with insomnia to decrease insomnia symptoms [19,20,21]. These results are of particular relevance for people struggling with comorbid IBS and insomnia, Since, comorbid IBS and insomnia are often characterized by high levels of pre-sleep cognitive arousal and pre-sleep somatic arousal [22], GI symptoms during sleep [6,23], maladaptive sleep habits and sleep-related dysfunctional cognitions [6].

Influenced by these studies, we investigate the effect of CBT-I, the interventional program that especially focuses on the characteristics of IBS, on both insomnia and IBS in college students with comorbid IBS and insomnia. The primary hypothesis of this study is that the severity of insomnia would decrease in the college students with comorbid IBS and insomnia who participated in CBT-I (the experimental group) compared to the college students with comorbid IBS and insomnia who did not participate in CBT-I (the control group). The secondary hypothesis is that the experimental group would have demonstrated levels in the following variables compared to the control group: pre-sleep arousal, sleep-related dysfunctional cognitions, maladaptive sleep habits, sleeping patterns, GI symptoms during sleep, inflammation (IL-6, CRP), severity of IBS symptoms and IBS quality of life.

## 2. Materials and Methods

### 2.1. Participants

College students who met the inclusion criteria were recruited through advertisements on the bulletin boards of two universities in Daegu, South Korea. The inclusion criteria were as follows: (1) male and female college students who were at least 18 years old; (2) meeting the criteria of Rome III Diagnostic Criteria for IBS [1]; (3) not diagnosed with any GI tract disorders such as inflammatory bowel disorders, lactose malabsorption and obstructive bowel disorders and no history of surgeries in the digestive system; (4) having insomnia (Insomnia Severity Index score ≥ 10) [24]; (5) not diagnosed with sleep apnea and shift work disorder; (5) not taking any medication that affected sleep or insomnia treatment (e.g., sleeping pills, antidepressants, antihistamines, etc.); (6) experiencing GI symptoms during sleep (e.g., abdominal pain, abdominal gas, defecation desire) at least once a week; and (7) not undergoing any treatment for infectious diseases (e.g., respiratory infections, GI tract infections, dental infections, etc.). Based on a preceding study [24], insomnia was defined by the insomnia severity index (ISI) score of 10 or more, and this ISI score is based on a response questionnaire that is answered by recalling sleep difficulties of the past month.

### 2.2. Randomization

A total of 60 college students who had met the inclusion criteria and agreed to participate in the research were given ID numbers. The participants were then randomly assigned either to the experimental or the control group (30 participants for each group) through a randomizer program (http://www.randomizer.org/form.htm, accessed on 10 November 2018) by a researcher who did not take part in data collection (Figure 1).

### 2.3. Sample Size Calculation

The sample size was calculated by applying the G*power 3.1 program to effect size 0.80, significance level 0.05 and power of test 0.80 for the independent *t*-test. Based on the finding of a meta-analysis that investigated the effect of cognitive behavioral therapy (CBT) on comorbid insomnia patients [25], the effect size was back-calculated to be 1.24 from the insomnia severity score (the experimental group: 9.61 ± 5.13, the control group: 15.91 ± 4.95).

However, we used 0.80, which was the highest effect size suggested by Cohen [26], because it was not the calculated result from IBS patients. Considering the dropout rate of 30%, 60 participants were required for the experimental and control groups.

A total of 222 participants were screened, 114 of which were excluded because they did not meet the Rome III criteria for IBS, and 48 participants were excluded because they scored below 10 points on the Insomnia Severity Index. The 60 participants who met the inclusion criteria all agreed to participate in the study and were randomly assigned to the experimental group (n = 30) and the control group (n = 30). During the study, one participant declined the intervention due to schedule conflict with his/her part-time job, thus a total of 59 participants, 29 from the experimental group and 30 from the control group, participated in and completed the study.

### 2.4. Intervention

Based on Spielman’s chronic insomnia 3-P model [27], CBT-I was developed for participants with comorbid IBS and insomnia to improve pre-sleep hyper-arousal, GI symptoms during sleep, sleep-related dysfunctional cognitions and maladaptive sleep habits. To this end, we reviewed the previous studies on CBT-I, analyzed the participant surveys, and investigated the experience of GI symptoms during sleep and patients’ needs through focus group interviews. For the detailed intervention contents of CBT-I, we referred to “Cognitive Behavioral Treatment of Insomnia: a session-by-session guide” [28] and “Case Study Book of Cognitive Behavioral Therapy for Insomnia” [29] and included a pain-related intervention for insomnia patients with chronic pain [30]. The final version of CBT-I was validated by experts and revised accordingly. The program was validated by an expert group consisting of two nursing professors and one psychiatrist who implements a cognitive behavioral therapy for insomnia in clinical practice. The content validity of the intervention was verified by using the Content Validity Index [CVI] evaluation table for adequacy, applicability and effectiveness.

With regard to the method of intervention, a CBT-I expert conducted face-to-face interventions in groups of 4–6 participants at a time. These sessions lasted for 90 min each and took place 4 times in 7-day intervals, based on the CBT-I study [30,31]. Before each session, the participants were given handouts for ring binders that contained the description of each session and what to keep in mind. The themes of each session were as follows. Session 1: Understanding and improving the environment and behaviors that disturb sleep; Session 2: Improving the thoughts and behaviors that disturb sleep; Session 3: Improving the thoughts and behaviors about GI symptoms that disturb sleep; and Session 4: Establishing plans for preventing recurrence and recapping the program. The details of the program are presented in Table 1. At the end of the study, the control group was provided with a booklet summarizing the contents of the CBT-I used in the experimental group and participated in a question and response session with the researcher.

### 2.5. Primary Outcomes

#### Insomnia Severity

Insomnia severity was measured using the Insomnia Severity Index (ISI), which consists of 7 items with a five-point Likert scale [32]. The ISI measures the severity of insomnia based on the criteria of the Diagnostic and Statistical Manual of Mental Disorders-Ⅳ (DSM-Ⅳ) and International Classification of Sleep Disorders (ICSD). The scores range from 0 to 28 points, and higher scores indicate higher levels of insomnia severity. The cases with over 10 points were determined to be suffering from insomnia [24]. The reliability of ISI at the time of development was Cronbach’s α = 0.74 and 0.87 for this study.

### 2.6. Secondary Outcomes

#### 2.6.1. Sleeping Pattern

In this study, we used a Korean version of the weekly sleep diary [28]. To examine the sleeping patterns, we analyzed the sleep diary entries in terms of Sleep Onset Latency (SOL), Wake Time after Sleep Onset (WASO), Total Sleep Time (TST), Total Time in Bed (TIB), average of Sleep Efficiency (SE). TST was calculated by subtracting SOL and WASO from TIB. SE was calculated by dividing TIB with TST and multiplying by 100 [28].

#### 2.6.2. GI Symptoms during Sleep

We used the question that Yang and Jun [6] developed based on the study by Rajbari et al. [23]. The question is about the frequency of sleep disturbance during a week due to abdominal pain, abdominal gas, or defecation desire. The study participants were asked to keep a daily record of the pain while keeping the sleep diary.

#### 2.6.3. Pre-Sleep Arousal

To measure the level of pre-sleep arousal, we used the Pre-Sleep Arousal Scale (PSAS) [33]. A total of 16 questions with a five-point Likert scale included 8 questions each for pre-sleep cognitive arousal and pre-sleep somatic arousal. The scores for each variable ranged from 8 to 40 points and higher scores indicated a higher level of pre-sleep arousal. The reliability of PSAS at the time of development was Cronbach’s α 0.77 and 0.89 for this study.

#### 2.6.4. Sleep-Related Dysfunctional Cognitions

To measure the sleep-related dysfunctional cognitions, we used the Dysfunctional Beliefs and Attitudes about Sleep Scale-16 (DBAS-16) [34], which consists of 16 items. Using the Visual Analogue Scale (VAS), the items were scored from 0 to 10 points. Higher scores indicated a high level of dysfunctional cognitions and attitudes related to sleep. The reliability of DBAS-16 at the time of development was Cronbach’s α 0.74 and 0.94 for this study.

#### 2.6.5. Maladaptive Sleep Habits

The Sleep Hygiene Practices Scale (SHPS) was used to measure the maladaptive sleep habits [35]. The SHPS consists of 30 items concerning arousal-related behaviors, sleep schedule, food and drink intake behavior and sleep environment. The questions were scored based on a six-point Likert scale, and the scores ranged from 30 to 180. Higher scores indicated inadequate sleep hygiene practices. The reliability of SHPS at the time of development was between 0.67 and 0.82 in subtypes. The reliability of the tool was 0.91.

#### 2.6.6. Inflammation

To measure the inflammation level, 5 cc of blood was drawn from each participant’s forearm between 8 A.M. and 10 A.M. after an 8 h fast. The collected blood specimen was immediately put in the EDTA tube treated with citric acid, shaken over 20 times to prevent clotting, and transported in an icebox to a laboratory. Then, it was centrifuged for 15 min at 3000 rpm, and the separated plasma supernatant was moved into 2.0 mL micro-tubes and kept at −70℃. IL-6 and CRP were measured using Microplate Reader (VERSA Max, USA) following the method of Enzyme-Linked Immunosorbent Assay (ELISA). The reagents used at the time of analysis were Quantikine ELISA Human IL-6 Immunoassay kit (R&D, Torrance, CA, USA) and Quantikine ELISA Human C-Reactive Protein/CRP Immunoassay kit (R&D, USA).

#### 2.6.7. Severity of IBS Symptoms

The severity of IBS symptoms was measured using the Irritable Bowel Syndrome-Severity Scoring System (IBS-SSS) developed by Francis et al. [36]. It consists of 7 items, and after excluding the categorical questions inquiring about the presence of abdominal pain or abdominal bloating, the remaining 5 questions were scored 100 points each, using the Visual Analogue Scale (VAS). The severity was classified as follows: 75–174 = mild, 175–299 = moderate, and 300–500 = severe. The reproducibility of IBS-SSS at the time of development was reported to be stable (85%), and the reliability in this study was Cronbach’s α 0.72.

#### 2.6.8. IBS Quality of Life

To measure the IBS quality of life, we used the Irritable Bowel Syndrome-Quality of Life (IBS-QOL) [37]. This test consists of 34 items with a five-point Likert scale. Higher scores indicate a higher quality of life. Cronbach’s α at the time of tool development was 0.95, and 0.96 for this study.

#### 2.6.9. Statistical Analysis

The collected data were analyzed using SPSS 22.0. The general characteristics of the participants were analyzed in percentage, means, and standard deviation. To verify the normal distribution, Shapiro–Wilk Test was performed. Regarding the general characteristics of the experimental and control groups, the homogeneity was tested using independent *t*-test, Chi-square test, and Fisher’s exact test. The homogeneity of the dependent variables of the two groups was pre-tested and analyzed using Mann–Whitney U-test or independent *t*-test. The hypotheses were tested and analyzed using independent *t*-test, Mann–Whitney U-test and repeated measured ANOVA.

### 2.7. Ethical Considerations

The present study was performed after obtaining the permission from the Institutional Review Board (IRB) of [details omitted for double-anonymized peer review]. All participants were provided with the details about the research objective, purpose, and method before they signed a written consent.

## 3. Results

### 3.1. General Characteristics of Participants and the Pre-Test for Homogeneity

The data analysis of the study participants showed that 88.1% (n = 52) of the participants were women, and among the IBS subtypes, the mixed type was the highest, 91.5% (n = 54) and their average age was 20 (Table 2). Among the dependent variables on the pre-test, the variables of sleeping pattern (i.e., sleep onset latency, wake after sleep onset, total sleep time, GI symptoms during sleep, and inflammation) did not show a normal distribution, while the remaining variables showed normal distribution. On the pre-test, the general characteristics and the dependent variables of both the experimental and control groups did not show any significant differences and confirmed the homogeneity in all categories.

### 3.2. Primary Hypothesis Test

#### Insomnia Severity

The insomnia severity of the experimental group decreased from 14.35 ± 3.37 on the pre-test to 6.59 ± 3.96 immediately after the intervention and 4.55 ± 3.43 three months after the intervention. In contrast, the control group did not show a significant change in insomnia severity, testing at 15.83 ± 3.37 on the pre-test, 14.63 ± 5.91 immediately after the intervention, and 15.27 ± 3.44 three months after the intervention. Statistical significance was also shown in the following categories: in between-group variation (F = 70.94, *p* < 0.001), time variation (F = 46.02, *p* < 0.001), and interaction between time and the groups (F = 32.84, *p* < 0.001) (Figure 2a).

### 3.3. Secondary Hypothesis Test

#### 3.3.1. Sleeping Patterns and GI Symptoms during Sleep

The SOL of the experimental group decreased by 24.76 ± 21.07 after the intervention, whereas the control group’s sleeping pattern increased by 4.87 ± 28.63, showing a significant difference between the groups (z = −4.15, *p* < 0.001) (Figure 3a). In addition, TIB also decreased by 15.59 ± 56.09 in the experimental group, whereas TIB of the control group increased by 21.70 ± 60.92, showing a significant difference between the groups (t = −2.44, *p* = 0.018) (Figure 3d). With regard to SE, the experimental group showed an increase by 7.41 ± 5.48, while the control group increased by 0.90 ± 6.59, showing a significant difference between the groups (t = −4.12, *p* < 0.001) (Figure 3e). Furthermore, the WASO of the experimental group decreased by 8.21 ± 16.34. The control group’s WASO also decreased by 3.60 ± 16.95; there was no significant difference between the groups (z = −1.77, *p* = 0.077) (Figure 3b). With regard to TST, the experimental group increased by 16.83 ± 61.01 and the control group also increased by 20.47 ± 60.54, showing no significant difference between the groups (z = −0.49, *p* = 0.628) (Figure 3c). With regard to GI symptoms during sleep, the experimental group decreased by 2.90 ± 3.22 after the intervention, while the control group decreased by 0.07 ± 2.26, showing a significant difference between the groups (z = −3.79, *p* < 0.001) (Figure 3f).

#### 3.3.2. Pre-Sleep Arousal

The pre-sleep cognitive arousal level of the experimental group decreased from 24.45 ± 6.04 on the pre-test to 14.90 ± 6.70 immediately after the intervention and 13.76 ± 6.28 three months after the intervention. By contrast, there was no significant change in the control group, which tested at 23.57 ± 6.29 on the pre-test, 24.47 ± 7.17 immediately after the intervention, and 26.30 ± 6.05 three months after the intervention, showing a statistical significance in both between-group variation (F = 35.96, *p* < 0.001), time variation (F = 17.53, *p* < 0.001), interaction between time and the groups (F = 37.71, *p* < 0.001) (Figure 2b). The pre-sleep somatic arousal of the experimental group decreased from 21.86 ± 6.10 on the pre-test to 14.14 ± 4.38 immediately after the intervention and 12.07 ± 3.54 three months after the intervention. By contrast, the control group did not show a significant change, testing at 22.33 ± 4.90 on the pre-test, 23.40 ± 6.69 immediately after intervention, and 24.20 ± 7.11 three months after the intervention, showing statistical significance in between-group variation (F = 38.58, *p* < 0.001), time variation (F = 15.80, *p* < 0.001), interaction between time and the groups (F = 32.18, *p* < 0.001) (Figure 2c).

#### 3.3.3. Sleep-Related Dysfunctional Cognitions

The sleep-related dysfunctional cognitions of the experimental group decreased from 89.00 ± 20.68 on the pre-test to 37.38 ± 23.22 immediately after the intervention and 26.48 ± 18.72 three months after the intervention. By contrast, the control group did not show a significant change, testing at 89.30 ± 18.66 on the pre-test, 85.47 ± 28.71 immediately after intervention, and 94.90 ± 19.58 three months after the intervention, showing a statistical significance in between-group variation (F = 65.87, *p* < 0.001), time variation (F = 73.81, *p* < 0.001), interaction between time and the groups (F = 85.73, *p* < 0.001) (Figure 2d).

#### 3.3.4. Maladaptive Sleep Habits

Maladaptive sleep habits of the experimental group decreased from 92.48 ± 18.47 on the pre-test to 59.69 ± 21.93 immediately after the intervention and 56.72 ± 18.412 three months after the intervention. By contrast, the control group did not show a significant change, testing at 91.40 ± 15.45 on the pre-test, 95.27 ± 20.35 immediately after intervention, and 97.63 ± 22.97 three months after the intervention, showing a statistical significance in between-group variation (F = 34.43, *p* < 0.001), time variation (F = 23.39, *p* < 0.001), interaction between time and the groups (F = 42.92, *p* < 0.001) (Figure 2e).

#### 3.3.5. Inflammation

IL-6 level of the experimental group decreased by 0.24 ± 1.47 pg/mL immediately after the intervention. Although the IL-6 level of the control group increased by 0.04 ± 1.00 pg/mL, there was no statistical difference between the groups (z = −1.60, *p* = 0.058) (Figure 3g). The CRP level of the experimental group did not show a noticeable change immediately after the intervention and the CRP level of the control group increased by 0.03 ± 0.15 mg/dL, showing no significant difference between the groups (z = −0.91, *p* = 0.363) (Figure 3h).

#### 3.3.6. Severity of IBS Symptoms

The severity of IBS symptoms in the experimental group decreased from 320.50 ± 63.18 on the pre-test to 193.03 ± 71.64 immediately after the intervention and 147.31 ± 89.89 three months after the intervention. By contrast, the control group did not show any significant change, testing at 312.52 ± 60.56 on the pre-test, 325.50 ± 46.12 immediately after the intervention, and 299.27 ± 85.57 three months after the intervention, showing a statistical significance in between-group variation (F = 45.33, *p* < 0.001), time variation (F = 38.12, *p* < 0.001), interaction between time and the groups (F = 32.84, *p* < 0.001) (Figure 2f).

#### 3.3.7. IBS Quality of Life

IBS quality of life in the experimental group increased from 126.48 ± 25.948 on the pre-test to 151.28 ± 15.33 immediately after the intervention and 161.55 ± 7.10 three months after the intervention. By contrast, the control group did not show a significant change over time, as they tested at 134.73 ± 21.70 on the pre-test, 132.63 ± 23.51 immediately after the intervention, and 121.80 ± 27.19 three months after the intervention, showing a statistical significance in between-group variation (F = 30.37, *p* < 0.001), time variation (F = 8.80, *p* < 0.001), interaction between time and the groups (F = 15.34, *p* < 0.001) (Figure 2g).

## 4. Discussion

The present study demonstrates the effect of CBT-I on the characteristics and symptoms of comorbid IBS and insomnia. The intervention decreased the severity of insomnia, pre-sleep arousal, GI symptoms during sleep, sleep-related dysfunctional cognitions, maladaptive sleep habits, IBS symptoms, IBS-QOL among college students with comorbid IBS and insomnia, while improving sleep patterns.

The Insomnia Severity Index of the experimental group decreased from 14.35 points in the pre-test to a normal range (4.55 points) after the intervention. Due to a lack of previous studies implementing CBT-I among IBS patients, a direct comparative analysis of past and present results is not possible. However, there are many studies supporting the effect of CBT-I on reducing the severity of insomnia among insomnia patients with chronic pain [19,30], osteoarthritis patients [20], and cancer patients [38]. In addition, there are also reports that reinforce the idea that interventions focusing on the characteristics and symptoms of comorbid conditions that cause insomnia can lead to the speedy treatment of insomnia and prevent its recurrence [39]. For example, in a study targeting osteoarthritis patients [20], compared to regular CBT-I, the Insomnia Severity Index decreased much more noticeably when the intervention was implemented for a particular characteristic and symptom of the disease.

The effect of the CBT-I intervention on the sleep pattern can be observed through the reduced SOL and TIB scores, which ultimately improved the SE of the college students with comorbid IBS and insomnia. However, improvement in SE did not result in significant changes in the TST. This could be due to the design of the intervention. On the second week of the CBT-I, we used a sleep compression method whereby we restricted the TIB and increased the sleep compression until the SE improved through sleep homeostasis before gradually increasing sleep time. In a study by Espie et al. [40], no significant changes in the TST were observed two weeks after implementing the sleep restriction method for people with insomnia, but there was a significant increase in the TST 12 months after the intervention. Given this, it is necessary that future studies measuring the effects of intervention set the time for their post-test much later. On the other hand, there was no significant decrease in the WASO onset after completing the intervention; however, the WASO of the experimental group decreased by 55% from 14.90 min on the pre-test to 6.69 min on the post-test. According to Morin et al. [41], a decrease of WASO by 30 min or by over 50% is an indicator for improvement in the sleeping pattern. Therefore, it can be concluded that the overall sleeping pattern in the present study improved.

It is speculated that GI symptoms during sleep occur because of the increased intra-abdominal pressure. The activation of GI motility occurring naturally during the rapid eye movement (REM) sleep can be perceived as abdominal pain or discomfort when the intra-abdominal hypersensitivity increases [6]. In our study, the GI symptoms during sleep improved. The evening diet education and pre-sleep relaxation training may have reduced the visceral perception that occurs during REM sleep by alleviating the intra-abdominal hypersensitivity and intestinal gas. To measure the GI symptoms during sleep, the present study asked participants to update their sleep diary, as soon as they wake up in the morning and record the number of times, they perceived GI symptoms during their sleep. However, many participants reported that it was difficult to recall their GI symptoms during sleep. Therefore, it is recommended that future studies consider adopting a tool that can objectively measure the GI symptoms during sleep.

CBT-I can help reduce pre-sleep arousal among the college students with comorbid IBS and insomnia. It is known that excessive pre-sleep arousal is a main factor that hinders sleep onset and causes insomnia. In previous studies [42,43], the effect of insomnia intervention on reducing the pre-sleep cognitive arousal have been reported; however, their effect on reducing the pre-sleep somatic arousal has not been discussed. The relaxation therapy, abdominal massage, and heat therapy that was conducted in this study seemed to have loosened the tension of abdominal and visceral muscles and activated the parasympathetic nerve and visceral nerve [44]. The therapy for dysfunctional cognitions caused due to GI symptoms may have alleviated the GI symptoms by suppressing the activation of the sympathetic nerve and visceral nerve causing anxiety and cognitive distortion [31,45]. As for patients with comorbid IBS and insomnia, pre-sleep somatic arousal often manifested as GI symptoms; and the strategy used for the pre-sleep GI symptoms could be directly responsible for reducing the pre-sleep somatic arousal.

In this study, CBT-I was effective in reducing sleep-related dysfunctional cognitions. This finding is similar to previous studies that reported the improvement in sleep-related dysfunctional cognitions by applying cognitive therapy for sleep-related dysfunctional cognitions targeting insomnia patients with comorbid restless leg syndrome [46] and chronic pain [30]. In addition, CBT-I was effective in reducing the maladaptive sleep habits. This finding is similar to a previous study that reported the alleviation of maladaptive sleep habits by applying CBT-I to patients with restless leg syndrome and by adding the method of sleep compression and the sleep hygiene education tailored to the characteristics of comorbid diseases [46].

The CBT-I in this study was also effective in reducing the severity of IBS symptoms. The CBT-I in this study was primarily used to treat insomnia, but it was also helpful in reducing the GI symptoms of IBS patients by inducing relaxation of the visceral pain caused due to abdominal hypersensitivity. Our study replicates the results of Jang et al. [31], where implementation of cognitive behavioral programs reduced GI symptoms and improved quality of life for people struggling with IBS.

Based on our findings, the following speculation can be made about the participants in this study. Through the relaxation and cognitive behavioral therapy for dysfunctional cognitions about pre-sleep GI symptoms, the participants must have experienced the effect of the program after they recognized that their distorted perceptions on GI symptoms can cause negative emotions and manifest as GI symptoms and dysfunctional behaviors. Then, they may have applied their new knowledge to sleep-related GI symptoms and to the thoughts and behaviors about the daytime GI symptoms. In fact, the in-depth interviews with three participants from the CBT-I group after the end of the program confirmed this speculation. All of them testified that they recognized the impact of their thinking and negative emotions on the manifestation of GI symptoms, and they utilized this knowledge for improving the daytime GI symptoms. They also reported about practicing abdominal breathing and experiencing the alleviation of GI symptoms under stressful situations.

Moreover, the CBT-I in this study was found to improve the participants’ quality of life. The quality of life of IBS patients is negatively correlated with the severity of their IBS symptoms, and abdominal pain was reported to exert the greatest impact on the quality of life [47]. Earlier in this study, it was explained that CBT-I can alleviate the hypersensitivity to visceral pain and improve the IBS severity by improving insomnia. Accordingly, the experimental group of this study may have improved their quality of life by applying the program to improve their abdominal pain.

On the other hand, the CBT-I in this study did not show any impact on the inflammation level among the participants. Inflammation is an essential immune response to maintain the homeostasis of infected areas or organs under various hazardous conditions. However, when insomnia persists, the products of the inflammation response including IL-6 and cytokine such as CRP remain concentrated in the blood [48]. The level of IL-6 in this study seemed to decrease immediately after the intervention in the experimental group, but there was no statistical difference. This result is similar to a previous study that showed an insignificant reduction in IL-6 level among the peritoneal dialysis patients after conducting 4 sessions of CBT-I [49]. There was also no significant change in CRP level immediately after the intervention. By contrast, in a study that conducted CBT-I for 16 weeks among the elderly patients with comorbid insomnia, there was a reduction in the CRP levels after the intervention [50]. It appears that the change in CRP level is hard to determine within a short period of time, considering the fact that the period of intervention for the study was rather long (16 weeks). Another longitudinal study also found that if someone experiences insomnia for over 6 years, it is most probably related to the rise of CRP levels [12]. Additionally, because the blood sampling for the post-test was done during the flu vaccination season (from November 30th to December 4th), this could have affected the rise in the inflammation level, even though there were no surface symptoms of infections.

In interpreting the findings of the present study, there are some limitations to be acknowledged. First, the sleep diary and the GI symptoms during sleep for a week were not measured at the 3-month follow-up because the participants had complained about the inconvenience of this method. Additionally, the blood sampling was done only once after the intervention, so we failed to clearly identify the change in the inflammation level, which takes a long time to manifest.

Based on the above findings, some suggestions can be made. First, it is necessary for follow-up studies to minimize the inconvenience for the participants. One way to do this is to use Actigraphy for measuring the sleeping pattern and GI symptoms during sleep and for double-checking the study findings by using a tool for objective measurement. Second, it is necessary to investigate the changes in the inflammation response as an indicator of a long-term improvement of the sleep disorder. Third, it is necessary to conduct a study that repeats the experiment by considering the gender of college students with comorbid IBS and insomnia, the characteristics of each IBS subtype, the circadian rhythm, and the impact of shift work. Fourth, it is necessary to investigate the effects of an individualized approach that identify each participant’s understanding of the CBT-I sessions and closely track changes in outcomes such as sleeping patterns and IBS symptoms. Fifth, since CBT-I was found to be effective in improving GI symptoms as well as sleep problems in IBS subjects with insomnia for the first time in this study, it is necessary to conduct a study comparing different types of interventions in the future.

## 5. Conclusions

By applying CBT-I along with the intervention focusing on the characteristics and symptoms of comorbid diseases that cause insomnia, the present study proved the effects of the program on reducing the severity of insomnia, IBS symptoms, and improving the quality of life of the college students with comorbid IBS and insomnia. Because there were no other direct interventions for insomnia designed for people with IBS, the CBT-I intervention used in this study has a great significance as a new approach for effectively improving insomnia and alleviating IBS symptoms. Therefore, the CBT-I developed in this study can be applied in clinical or community settings to overcome insomnia, IBS symptoms and elevate the quality of life for people with comorbid IBS and insomnia.

## Figures and Tables

**Figure 1 ijerph-19-14174-f001:**
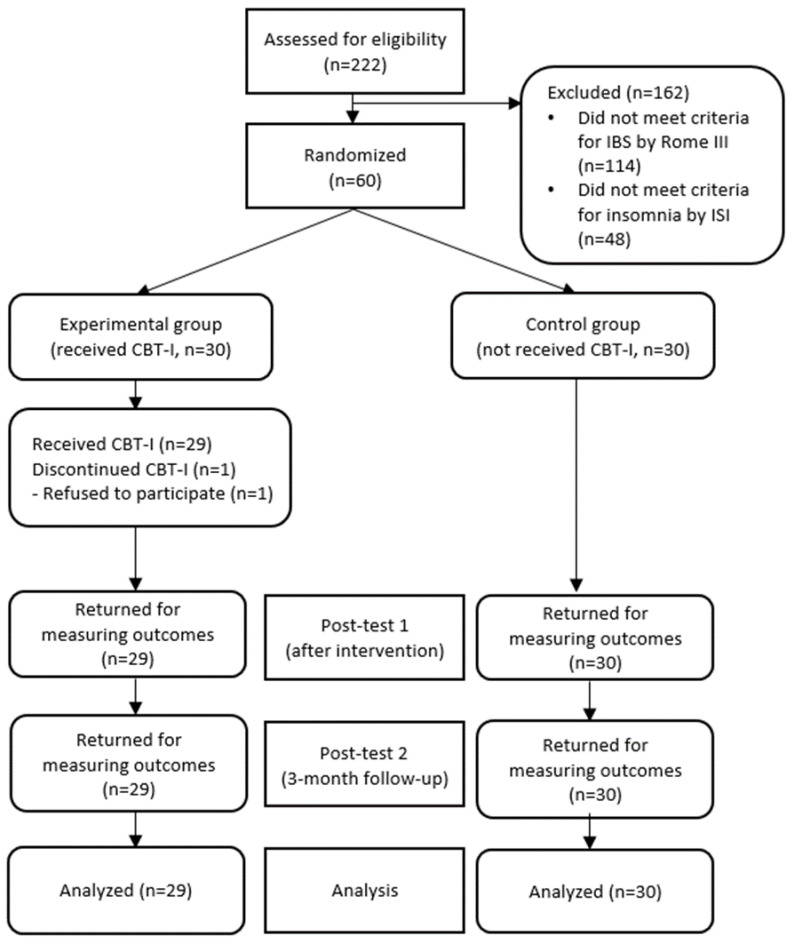
Flow diagram of inclusion and exclusion. ISI = insomnia severity index, CBT-1 = cognitive behavioral therapy for insomnia.

**Figure 2 ijerph-19-14174-f002:**
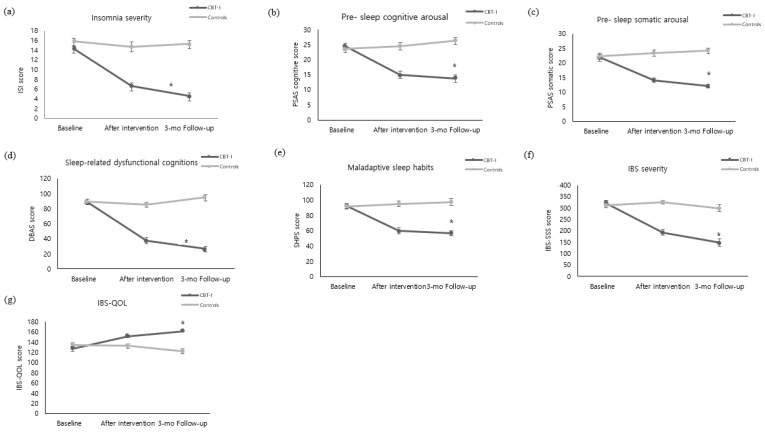
Comparison of outcome variables of the experimental and control groups over time ((**a**) Insomnia severity. (**b**) Pre-sleep cognitive arousal. (**c**) Pre-sleep somatic arousal. (**d**) Sleep-related dysfunctional cognitions. (**e**) Maladaptive sleep habits. (**f**) IBS severity. (**g**) IBS-QOL). The *p* value was calculated using repeated measured ANOVA. CBI-I: Cognitive behavioral therapy for insomnia. ISI: Insomnia Severity Index. PSAS: Pre-Sleep Arousal Scale. DBAS: Dysfunctional Beliefs and Attitudes about Sleep Scale. SHPS: Sleep Hygiene Practices Scale. IBS: Irritable Bowel Syndrome. IBS-QOL: Irritable Bowel Syndrome-Quality of Life. * *p* < 0.001.

**Figure 3 ijerph-19-14174-f003:**
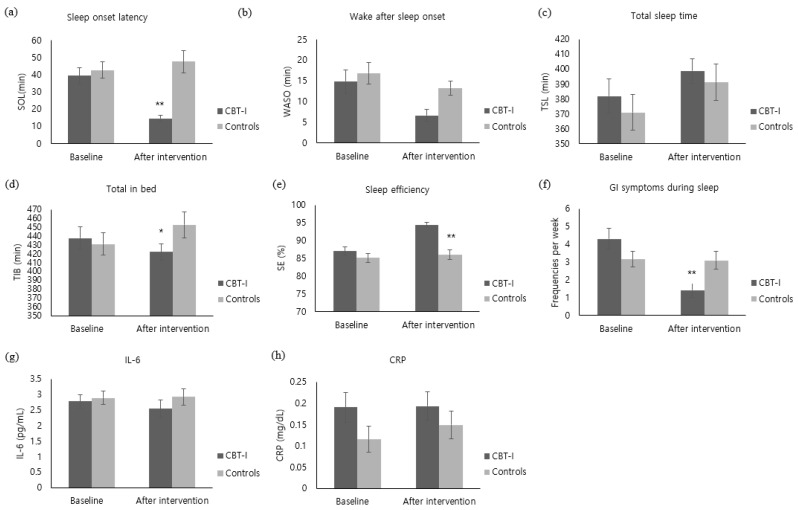
Comparison of outcome variables of the CBT-I and the control groups at baseline and after intervention ((**a**) Sleep onset latency. (**b**) Wake after sleep onset. (**c**) Total sleep time. (**d**) Total in bed. (**e**) sleep efficiency. (**f**) GI symptoms during sleep. (**g**) IL-6. (**h**) CRP). CBI-I: Cognitive behavioral therapy for insomnia. SOL: Sleep Onset Latency. WASO: Wake Time after Sleep Onset. TST: Total Sleep Time. TIB: Total Time in Bed. SE: Sleep Efficiency. GI: Gastrointestinal. IL-6: Interleukin-6. CRP: C-Reaction Protein. * Cognitive behavioral therapy for insomnia (CBT-I) group versus control group, *p* < 0.05. ** Cognitive behavioral therapy for insomnia (CBT-I) group versus control group, *p* < 0.001.

**Table 1 ijerph-19-14174-t001:** Details of the Cognitive Behavioral Therapy for Insomnia (CBT-I) for IBS subjects with insomnia.

Themes and Objectives	Contents
Session 1. Understanding and improving the environment and behaviors that disturb sleep	
Improving maladaptive sleep habits	▸ Sleep education for understanding insomnia and IBS
	▸ Sleep hygiene education for IBS
Improving GI symptoms during sleep	▸ Relaxation training for IBS
Session 2. Improving the thoughts and behaviors that disturb sleep	
Improving sleep-related dysfunctional cognitions	▸ Cognitive therapy for sleep-related dysfunctional cognitions
Improving maladaptive sleep habits	▸ Sleep compression and stimulus control for IBS
Session 3. Improving the thoughts and behaviors about GI symptoms that disturb sleep	
Improving pre-sleep hyper-arousal	▸ Cognitive therapy for dysfunctional cognitions about GI symptoms during sleep
	▸ Creating buffer zone and setting a time for worries for IBS
Improving the GI symptoms during sleep	▸ Evening diet education for IBS
Session 4. Establishing plans for preventing reoccurrence and recapping the program	
Improving the sleep-related dysfunctional cognitions, maladaptive sleep habits, GI symptoms during sleep, pre-sleep hyper-arousal	▸ Summary and identify sleep pattern change
	▸ Establishing the measure for reoccurrence and maintenance

IBS: irritable bowel syndrome; GI; gastrointestinal; FODMAP: Fermentable, Oligo-, Di-, Mono-, Saccharides and Polyols.

**Table 2 ijerph-19-14174-t002:** Baseline characteristics and homogeneity of participants (*n*= 59).

Characteristics	Categories	CBT-I (*n* = 29)	Controls (*n* = 30)	*χ*^2^/t	*p*
		*n* (%) or Mean ± SD	*n* (%) or Mean ± SD		
Age (years)		20.50 ± 3.31	20.42 ± 2.31	0.17	0.908 *
Gender	Male	3(10.3)	4(13.3)	0.13	1.00 ^†^
	woman	26(89.7)	26(86.7)		
IBS subtypes	Diarrhea-predominant	0(0.0)	2(6.7)	2.40	0.303 ^†^
	Constipation -predominant	1(3.4)	2(6.7)		
	Mixed	28(96.6)	26(86.7)		
Smoking	Yes	4(13.8)	6(20.0)	0.40	0.731 ^†^
	No	25(86.2)	24(80.0)		
Drinking	Yes	16(55.2)	11(13.7)	2.04	0.154 ^‡^
	No	13(44.8)	19(63.3)		
Sleep partner	Yes	6(20.7)	11(36.7)	1.84	0.252 ^‡^
	No	23(79.3)	19(63.3)		
Night duty	Yes	4(13.8)	11(36.7)	4.07	0.071 ^‡^
	No	25(86.2)	19(63.3)		
Insomnia severity		14.35 ± 3.37	15.83 ± 3.37	1.69	0.096
Sleeping pattern	Sleep onset latency (min)	39.48 ± 26.01	42.80 ± 26.80	−0.46	0.644 *
	Wake time after sleep onset (min)	14.90 ± 15.00	16.90 ± 14.49	−0.75	0.452 *
	Total sleep time (min)	382.00 ± 61.16	370.87 ± 65.58	−0.68	0.503
	Total time in bed (min)	437.48 ± 67.48	431.00 ± 67.72	−0.37	0.714
	Sleep efficiency (%)	87.07 ± 5.98	85.17 ± 6.96	−1.12	0.265
Pre-sleep arousal	Cognitive	24.45 ± 6.04	23.57 ± 6.29	−0.55	0.585
	Somatic	21.86 ± 6.10	22.33 ± 4.90	0.33	0.744
GI symptoms during sleep		4.31 ± 3.15	3.17 ± 2.36	−1.39	0.164 *
Sleep-related dysfunctional cognitions		89.00 ± 20.68	89.30 ± 18.66	0.06	0.954
Maladaptive sleep habits		92.48 ± 18.47	91.40 ± 15.45	−0.25	0.808
Inflammation	IL-6 (pg/mL)	2.78 ± 1.13	2.89 ± 1.20	−1.02	0.310 *
	CRP (mg/dL)	0.19 ± 0.19	0.12 ± 0.17	−1.75	0.080 *
IBS severity		320.52 ± 63.18	312.50 ± 60.56	−0.50	0.621
IBS QOL		126.48 ± 25.94	134.73 ± 21.70	1.33	0.190

CBT-I: Cognitive Behavioral Therapy for Insomnia; IBS: irritable bowel syndrome; GI; gastrointestinal; IL-6: Interleukin-6; CRP: C-Reaction Protein; QOL: quality of life; * independent *t*-test; ^†^ Fisher’s exact test; ^‡^ Chi-square test.

## Data Availability

The data presented in this study are available on request from the corresponding author. However, to maintain confidentiality, the data are not publicly available.

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
