# Peer review of "The Effects of Cognitive Behavioral Therapy for Insomnia among College Students with Irritable Bowel Syndrome: A Randomized Controlled Trial"

_ijerph, 2022, doi:10.3390/ijerph192114174_

Round 1

Reviewer 1 Report

The manuscript by Yang et al., 2022 present the effect of cognitive behavioral therapy on insomnia among student with irritable bowel syndrome. In the study they randomly as-signed 60 college student with IBS with insomnia. For the treatments they used CBT-1 (cognitive behavioral therapy) which contained of 90 min once a week for 4 weeks. In this CBT-1 session patient were thought different methods they could use to improving their sleeping habits.

I liked the approach of the study and to measure the impact of this noninvasive method on patients, where people were advising how they can with their own work on their problems try to find a solution for it. With this they can eliminate the source of the problems such as insomnia not just healing the consequences of the insomnia by using different treatments such as medicines.

For the future I still see some chances of improvement the methodology to get better result.

Individual approach and tracking each patient individual would improve the outcome, in a way that doctor would have individual meeting with the patient after the sessions about what he learned at the session, how he his health is improving with the questionary, what he understand/not understanding, what he thinks could be improved……also to be collected how sever is they IBS disease. The difference between CBT-1 treatment and untreated it not huge in table one. This could be improved with more specific work with individual. In figure 2 and 3 the X-axis and Y-axis are not labeled with numbers properly. It looks like someone wants to emphasis the difference without labeling the values properly in the graphs.

Author Response

We appreciate the careful review given to our manuscript and address the following concerns below:

Reviewer #1

  1. Individual approach and tracking each patient individual would improve the outcome, in a way that doctor would have individual meeting with the patient after the sessions about what he learned at the session, how he his health is improving with the questionary, what he understand/not understanding, what he thinks could be improved……also to be collected how sever is they IBS disease.

Response: We appreciate for suggesting a more in-depth methods that can be applied in the clinical setting. The following suggestions have been added in the discussion section:   

“Fourth, it is necessary to investigate the effects of an individualized approach or program that identify each participant’s understanding of the CBT-I sessions and closely track changes in outcomes such as sleeping patterns and IBS symptom severity.”

  1. The difference between CBT-1 treatment and untreated it not huge in table one. This could be improved with more specific work with individual.

Response: Table 1 presents the details of the CBT-1 program. If the table 1 mentioned in the reviewer’s comments refers to Table 2, Table 2 presents the baseline data indicating the homogeneity of the control and experimental groups in the general and disease-related characteristics before the intervention started.  

  1. In figure 2 and 3 the X-axis and Y-axis are not labeled with numbers properly. It looks like someone wants to emphasis the difference without labeling the values properly in the graphs.

Response: Thank you for your kind reminders. According to reviewer’s comments, the Y-axis labels of each graph were properly revised starting at 0 in Figure 2 and 3; however, several variables such as total sleep time or total time in bed were truncated because the numbers were too large. The X-axis represents the time points of data collection.

Reviewer 2 Report

The inclusion of a control group in which no type of intervention similar to that of the study group is performed may establish a risk of bias since in the control group there is an effect of the CBT sessions. This type of design is considered to be of lesser value than one in which two types of intervention are compared. Seen this way, an intragroup analysis would be missing for each of the study groups with tests for related samples.The inclusion of a control group in which no type of intervention similar to that of the study group is performed may establish a risk of bias since in the control group there is an effect of the CBT sessions. This type of design is considered to be of lesser value than one in which two types of intervention are compared. Seen this way, an intragroup analysis would be missing for each of the study groups with tests for related samples.

Author Response

We appreciate the careful review given to our manuscript and address the following concerns below:

Reviewer #2

The inclusion of a control group in which no type of intervention similar to that of the study group is performed may establish a risk of bias since in the control group there is an effect of the CBT sessions. This type of design is considered to be of lesser value than one in which two types of intervention are compared. Seen this way, an intragroup analysis would be missing for each of the study groups with tests for related samples.

Response: We appreciate for reviewer’s kind comments and agree with him/her.

We did our best to prevent the spread of CBT-I sessions to the control group like using a blind method so that participants did not know each other to the control and experimental groups.

In addition, the purpose of this study was to investigate whether CBT-I for the IBS subjects with insomnia also had effects on the improvement of GI symptoms. For the first time in this study, CBT-I was found to be effective in improving not only insomnia but also GI symptoms. Therefore, it is necessary to conduct a study utilizing a design comparing two types of interventions as appropriate to future studies’ purposes in the future. The following suggestions have been added in the discussion section:   

“Fifth, since CBT-I was found to be effective in improving GI symptoms as well as sleep problems in IBS subjects with insomnia for the first time in this study, it is necessary to conduct a study comparing different types of interventions in the future”.